# Computer-based cognitive interventions in acquired brain injury: A systematic review and meta-analysis of randomized controlled trials

**Rodrigo Fernández López**[1]*, **Adoración Antolí**[1,2]

**1** Department of Psychology, University of Córdoba, Córdoba, Spain, **2** Maimónides Institute for Biomedical Research in Córdoba (IMIBIC), Reina Sofía University Hospital, University of Córdoba, Córdoba, Spain

☯ These authors contributed equally to this work.
* rodrigofdezlo@gmail.com

## Abstract

### Introduction

Acquired brain injury (ABI) leads to cognitive deficits in a great variety of cognitive functions. Interventions aimed at reducing such deficits include the use of computer-based cognitive interventions. The present work synthetizes and quantitively analyses the effect of computer-based cognitive interventions in ABI.

### Methods

PubMed, Scopus, Web of Science, ProQuest and Ovid databases were searched for randomized controlled trials (RCT) addressing this issue. A total number of 8 randomized-controlled trials were included for systematic review and meta-analysis. Univariate meta-analyses were conducted for every cognitive function, producing aggregates when a study contributed more than one effect size per cognitive domain.

### Results

Random-effects meta-analyses showed an improvement of Visual and Verbal working memory, while other domains like Attention, Processing speed, Executive functions and Memory were not benefited by the interventions.

### Conclusions

Computer-based cognitive interventions might be a beneficial intervention for ABI population to improve Visual and Verbal working memory, although no effect was found in other cognitive domains. Implications and possible future directions of the research are discussed.

**Data Availability Statement:** All the study data and R code are publicly available at: https://osf.io/

9r3ks/?view_only=
145ae9c0bbb8418796832ea1d73f28d0 and full
report of non-included analyses can be found in
Supporting Information files within the manuscript.

**Funding:** This work was supported by funds from
the Andalusian Regional Government (Spain) (PIN-
0072-2016), (UCO-FEDER 18 REF. 1265277 MD
A1), and the University of Córdoba (Spain)
(PPG2018-UCOSOCIAL-11) to AA. The funders had
no role in study design, data collection and
analysis, decision to publish, or preparation of the
manuscript.

**Competing interests:** The authors have declared
that no competing interests exist.

# Introduction

Acquired brain injury (ABI) can lead to deficits in attention, memory, executive functions and processing speed. Approximately, 50–60% of traumatic brain injury (TBI) patients manifest memory and attention problems, and 30% of the patients require assistance in the activities of daily living [1]. On the other hand, between 70–96% of stroke patients have cognitive impairment to some degree [2]. These deficits can lead to a loss of functional independence [3] and disability [4]. ABI represents a major global health burden. Estimated costs for stoke and TBI were EUR 97.1 billion in Europe [5] and USD 221 billion in the United States [6].

Cognitive intervention refers to the provision of neuropsychological interventions aimed at rehabilitating, restoring or compensating neurocognitive impairments after ABI. The interest in cognitive rehabilitation has increased over the years [7] and there are considerable efforts in the search of evidence-based interventions [8] for ABI patients. Computer-based cognitive interventions are potentially an important tool for the rehabilitation of neurocognitive impairments. Theoretically, computer-based cognitive interventions can be sensitive to the user performance and adapt in real time the level of difficulty or the nature of the task. Additionally, they present advantages over classical cognitive interventions, allowing the standardization of the intervention and providing performance data to both the user and the professional, which can help to adapt the intervention to the patient's needs. Computer-based cognitive interventions are usually based on direct cognitive training of the different cognitive domains. Cognitive interventions can be multi-domain or single-domain. Multi-domain interventions target several different cognitive domains (e.g. memory, executive functions and working memory) throughout the duration of the intervention. On the other hand, single-domain interventions focus in the training of a single cognitive domain along the whole process. Some of these computer-based interventions can be carried out by the patient alone or with the therapist involved. Computer-based programs can be used in addition with other compensatory techniques such as strategy training in order to improve the performance of the patients in the given task.

Several meta-analyses have addressed the efficacy of computer-based cognitive interventions in Dementia [9, 10], Mild Cognitive Impairment (MCI) [11] and healthy population [12]. Although the trials included in these studies had methodological problems that could be improved, they showed a general positive small effect on cognition. Previous systematic reviews in mild traumatic brain injury [13], stroke [14] and ABI [15] population suggest that computer-based cognitive interventions might be effective, although the current available evidence is weak and methodologically flawed. To date, no meta-analysis has been performed on computer-based cognitive interventions for ABI.

Thus, the objective of this study is to systematically review and meta-analyze randomized controlled trials (RCT) that study the effect of computerized cognitive interventions for ABI in the different cognitive domains.

# Methods

This study was conducted following the recommendations of Preferred Reporting Items for Systematic Review and Meta-Analyses (PRISMA) guidelines [16] (see S1 Table). The review and meta-analysis protocol (#CRD42019138833) was pre-registered with PROSPERO International Prospective Register of Systematic Reviews. All the data and R code used in this study can be freely accessed and consulted at: https://osf.io/9r3ks/?view_only=
145ae9c0bbb8418796832ea1d73f28d0

## Eligibility criteria

Inclusion criteria were as follows [1] randomized-controlled trials, [2] active or passive control group, [3] acquired brain injury population, [4] intervention aimed at improving cognition [5] use of computer-based intervention, [6] evaluation of cognition using standardized tests immediately post-intervention and [7] pre-post design.

Exclusion criteria were as follows [1] non-randomized trials, [2] uncontrolled trials, [3] trials with inadequate control group (e.g. comparing standard cognitive intervention vs computer-based), [4] participants with neurodegenerative conditions, [5] participants with neurodevelopmental disorders, [7] absence of cognitive assessment, [8] testing was not done immediately after intervention and [9] insufficient data to estimate effect size.

## Information sources

We conducted a search in PubMed, Scopus, ProQuest, Ovid and Web of Science databases. We also consulted references from studies and accessed Google Scholar database. The latest search was carried out in September 2019 without any time or language restriction.

## Search strategy

The search equation applied to all the databases was: ("cognitive intervention" OR "cognitive training" OR "cognitive stimulation" OR "cognitive remediation" OR "cognitive rehabilitation" OR "neuropsychological intervention" OR "attention training" OR "memory training" OR "executive function* training" OR "flexibility training" OR "processing speed training" OR "working memory training" OR "brain training" OR "brain games" OR "reasoning training" OR "mental training" OR "neurocognitive training") AND ("computer-based" OR "computerized" OR "videogame" OR "computer game") AND ("acquired brain injury" OR "traumatic brain injury" OR"stroke" OR "brain injuries" OR "cerebrovascular disorders").

The search terms were developed collaboratively by the two review authors (RFL, Rodrigo Fernández López and AAC, Adoración Antolí Cabrera).

## Selection process

Eligibility was assessed by RFL following the exclusion and inclusion criteria. After the electronic search and duplicate removal, article titles and abstracts were screened. The full-text articles that met initial eligibility criteria were assessed for a final determination of eligibility by the two review authors independently. When a disagreement occurred, the decision was made by consensus. When there was insufficient information to assess eligibility (e.g. unclear control group), the authors were contacted for further information.

## Data collection process

A spreadsheet was created containing the main coded variables for the study. The items coded for each effect size were [1] demographics (e.g., %female, age), [2] type of ABI (e.g., stroke, traumatic brain injury, mixed), [3] intervention domain (e.g., working memory, memory, executive functions, processing speed, attention), [4] intervention duration, dosage and characteristics, [5] control group design (i.e., passive or active) and [6] neuropsychological assessment (e.g., digit span forward, trail making test, WAIS sub-tests). When necessary data to estimate effect size was missing, study authors were contacted via e-mail. When study authors did not answer within one month, a second e-mail was sent asking for the data. A total number of 6 authors were contacted via e-mail, of which 2 authors responded. Of those 2 authors, one

author was not able to access the data we were requesting, and the other author did not send us the data.

Intervention domains and outcome measures were coded according to previous works [17, 18] that categorize the data into cognitive domains based on the description of the intervention and assessment on each study.

## Summary measures

Effect sizes were calculated as standardized mean differences (SMD) with a 95% confidence interval (CI) using the Hedge's *g* estimator. To estimate effect size and variance, the formula for pre-post study designs with a control group proposed by Morris [19] was used. Positive values are interpreted as an improvement on the cognitive function of the experimental group compared to the control group. When an outcome measure reflected improvement by scoring less (e.g., reaction times), the *g* values were inverted. According to Cohen's criteria, values of 0.2–0.5 are interpreted as small effect, 0.5–0.8 as medium effect, and >0.8 as large effect.

## Synthesis of results

A random-effects model was assumed following the recommendations of Borenstein et al. [20] given the heterogeneity between the samples and the neuropsychological tests. The Sidik-Jonkman estimator of variance was used to conduct the random-effects model, since it has been found that it is a good estimator when heterogeneity variance is large [21].

When a study included more than one post-treatment measures, we only calculated the effect size for the measure closer in time to the end of the intervention. When there were more than one valid control group, we combined the groups into a single following the recommendations in the Chapter 7 of the Cochrane Handbook [22].

When a study reported multiple outcome measures for the same cognitive function, aggregates were generated using de Agg function of the MAD package in R [23] as suggested by Borenstein et al. [20] to avoid violating the assumptions of independence. The correlation between outcomes was set at 0.5 by default. For each cognitive function, univariate meta-analyses and forest plots were calculated using the metaphor package for R [24] and the *p*-value significance was set at < .05.

The heterogeneity of the effect sizes of the studies was assessed using the Cochrane's *Q* statistic, the $\tau^2$ and the $I^2$. $I^2$ was interpreted according to Higgins et al. [25], where $I^2$ = 25% means low heterogeneity, $I^2$ = 50% means medium heterogeneity and $I^2$ = 75% means high heterogeneity. Due to the low number of studies included in the analysis (k<10), funnel plot and meta-regression analyses were not calculated following the recommendations of the Cochrane Handbook [22, 26].

## Risk of bias in individual studies

Risk of bias was assessed using the Cochrane Risk of Bias Tool for experimental studies and the guidelines in Chapter 8 of the Cochrane Handbook [22] by RFL. When there was uncertainty in the coding of risk of bias, the two review authors decided by consensus.

## Results

### Study selection

Initial search in databases returned 3181 records and 3 records were identified from other sources. After removal of duplicates, 3095 unique records remained. These records were screened by RFL by reading the title and abstract, obtaining a total of 3048 excluded records

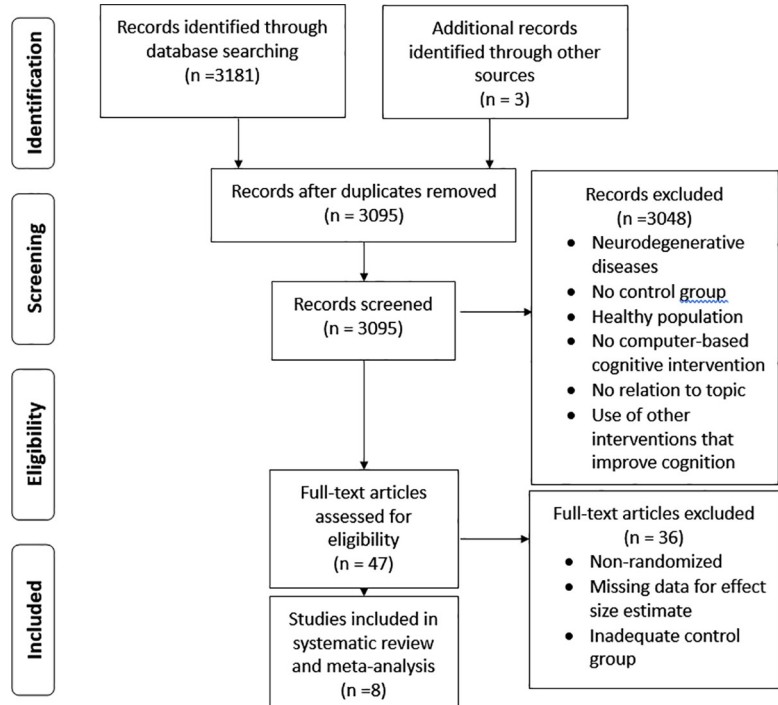

**Fig 1. PRISMA flowchart of the studies included in the systematic review and meta-analysis.**

and 37 included records to assess for full-text eligibility. After full-text assessing, 8 studies were included for quantitative analyses and systematic review. The selection process and the reasons for exclusion are summarized in the PRISMA flowchart in Fig 1.

## Study characteristics

Table 1 summarizes the study characteristics of the trials selected for the meta-analysis, including author, year, N, population, age, sex, intervention domain, intervention format,

**Table 1. Summary of study and intervention characteristics.**

| Author | Year | N | Population | Age | Sex% female | Intervention domain | Interventionformat | Interventionmethod | Control |
|---|---|---|---|---|---|---|---|---|---|
| Åkerlund et al. [27] | 2013 | 45 | ABI | 47.70 (11.27) | 49.0 | Working memory | Individual | Cognitive training (Cogmed) | Usual care |
| Cho, Kim & Jung [28] | 2015 | 25 | Stroke | 61.92 (5.78) | 36.0 | Attention | Individual | Cognitive training (RehaCom) | Usual care |
| Lin et al. [29] | 2014 | 34 | Stroke | 62.82 (5.77) | 41.2 | Executive function | Individual | Cognitive traning (Rehacom) | Passive group |
| Man et al. [30] | 2006 | 48 | ABI | 45.13 (19.96) | 38.0 | Problem-Solving | Individual | Cognitive training and strategy training | Passive group |
| Piovesana et al. [31] | 2017 | 57 | TBI | 11.88 (2.48) | 48.3 | Multi-domain | Individual | Cognitive training (Mitii) | Usual care |
| Van de Ven et al. [32] | 2017 | 97 | Stroke | 59.45 (8.67) | 30.7 | Multi-domain | Individual | Cognitive training (BrainGymmer) | Active and passive control group |
| Westerbeg et al. [33] | 2007 | 18 | Stroke | 54 (7.70) | 33.3 | Working memory | Individual | Cognitive training (RoboMemo) | Passive group |
| Yoo et al. [34] | 2015 | 46 | Stroke | 54.75 (8.40) | 63.0 | Multi-domain | Individual | Cognitive training (RehaCom) | Usual care |

**Table 2. Relevant characteristics of the computer-based cognitive interventions.**

| Author | Date | Duration of intervention (hours) | Frequency (Sessions/week) | Length of session (min) | Number of sessions | Setting | Time since injury (months) | Therapist during intervention |
|---|---|---|---|---|---|---|---|---|
| Åkerlund et al. [27] | 2013 | 15 | 5 | 30–45 | 25 | Outpatient | 33.18 (25.42) | No (feedback provided once a week) |
| Cho, Kim & Jung [28] | 2015 | 15 | 5 | 30 | 30 | Inpatient | 5.67 (2.23) | Yes |
| Lin et al. [29] | 2014 | 60 | 6 | 60 | 60 | Outpatient | 7.49 (0.69) | Yes |
| Man et al. [30] | 2006 | 15 | 1 | 45 | 20 | Home | 44.79 (47.19) | Yes |
| Piovesana et al. [31] | 2017 | 17.57 | NR | NR | NR | Home | NR | No (feedback provided once a week) |
| van de Ven et al. [32] | 2017 | 29 | 5 | 30 | 58 | Home | 28.52 (15.71) | No (feedback provided once a week) |
| Westerbeg et al. [33] | 2007 | 13.3 | NR | 40 | 23 | Home | 20.11 (6) | No (feedback provided once a week) |
| Yoo et al. [34] | 2015 | 12.5 | 5 | 30 | 25 | Inpatient | 11.26 (6.83) | NR |

intervention method, and type of control group. Other relevant characteristics of the interventions provided are detailed in Table 2.

The included studies resulted in a total of 370 participants (M = 46.25, SD = 24.19). There was heterogeneity in terms of age (M = 50.41, SD = 16.45), and there was general consistency in the sex of the participants (% females, M = 42.44, SD = 10.6). Stroke was the most common condition in the participants (k = 5), followed by mixed samples of TBI and Stroke (k = 2) and TBI alone (k = 1). In relation to the type of intervention, all the included studies provided individual cognitive interventions (k = 8). The targeted cognitive domains were different between trials, being the multi-domain interventions the most common ones (k = 4), followed by single-domain interventions in working memory (k = 2), attention (k = 1) and problem solving (k = 1).

Trials can use different types of control groups. Active control group refers to the provision of another intervention to the control group that is known to not affect the variables being assessed. On the other hand, passive groups do not participate in any kind of intervention. Additionally, there is another kind of control group in which the usual care of the setting is provided (e.g. Physical rehabilitation) without the intervention given to the experimental group. The most common control group was usual care (k = 4), followed by passive groups (k = 3) and active control group (k = 1).

Regarding the neuropsychological assessment, there was a high heterogeneity, with different studies providing several effect sizes for the same cognitive domain and several trials providing no effect size for some of them. See S1 Table for a detailed table containing all the tests used for every cognitive function by the studies included in the meta-analysis.

There was substantial variability in the duration of the interventions provided (hours, M = 22.17, SD = 15.11). While most trials provided interventions in the range of 10–20 hours (k = 6), others reached 29 hours (k = 1) and 60 hours (k = 1). The same happens with the number of sessions provided to the participants (M = 34.43, SD = 15.79), ranging from 20 to 30 sessions for the studies that provide less hours (k = 6) and 58 to 60 (k = 2) for longer ones In terms of frequency, the most common was 5 sessions per week (k = 4), followed by 6 sessions per week (k = 1) and 1 session per week (k = 1). Note that Piovesana et al. [28] and Westerberg et al. [30] do not report frequency because patients were able to self-manage the amount of sessions per week. Regarding the length of sessions (M = 38.93, SD = 10.16), most of the studies are in the range of 30–45 minutes (k = 6), while only one trial provides 60 minutes (k = 1). The

most common setting in which the interventions took place was at home (k = 4), while other trials were performed in an outpatient setting (k = 2) or in an inpatient setting (k = 2). In the trials included, 3 of them had a therapist providing the intervention, while in 4 studies the intervention was completed by the patient alone with weekly feedback from the therapist.

There was considerably variability in the time since injury presented by the patients (M = 23.36, SD = 26.56). The most usual time since injury is less than 12 months (k = 3). There were trials in the range of 12 to 24 months (k = 1) and 24 to 36 months (k = 2), up to more than 36 months (k = 1).

### Risk of bias in individual studies

The details of the risk of bias assessment can be seen in Fig 2 and Fig 3. Generally, a good methodological quality is assumed given that all the included studies are RCTs. However, the most common methodological flaw that the studies presented was a lack of the description of the randomization process and the absence of allocation concealment. Additionally, there was a lack of blinding for participants and neuropsychological assessors. Most trials did not report a planned protocol for the study (e.g. Pre-registration), which could lead to selective reporting of the outcome measures or deviations from the intended intentions. Due to the low number of studies included, moderator analyses based on the methodological quality of the studies could were not performed.

### Synthesis of results

Separate univariate meta-analyses were performed for every cognitive domain addressed by the studies. Forest plots of statistically significant cognitive domains are depicted in Fig 4 and Fig 5. The forest plots of non-significant outcomes can be seen at S1 File. In general, all the cognitive functions showed a positive effect of the cognitive intervention of the study participants, although not all of them reached statistical significance. The cognitive domains that showed a significant improvement were: Verbal working memory (SMD = 0.486, 95% CI [0.034, 0.938], p = 0.035, small effect) and Visual working memory (SMD = 0.543, 95% CI [0.189, 0.897], p < 0.01, medium effect).

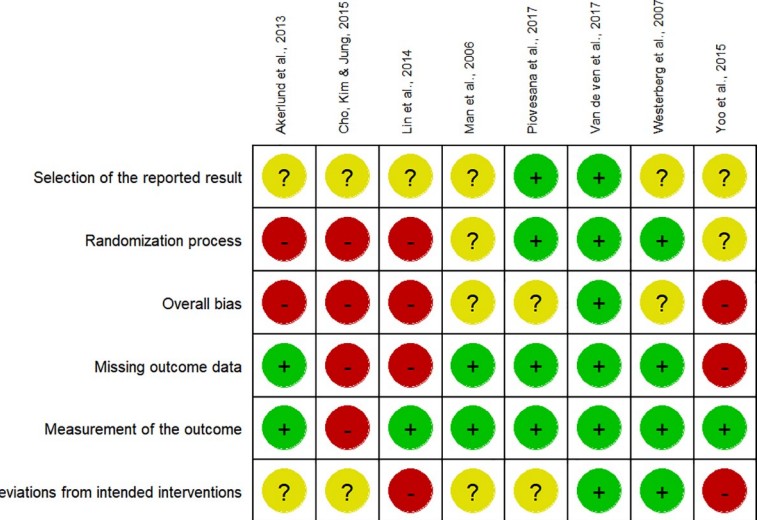

**Fig 2. Risk of bias assessment of individual studies included in the meta-analysis.**

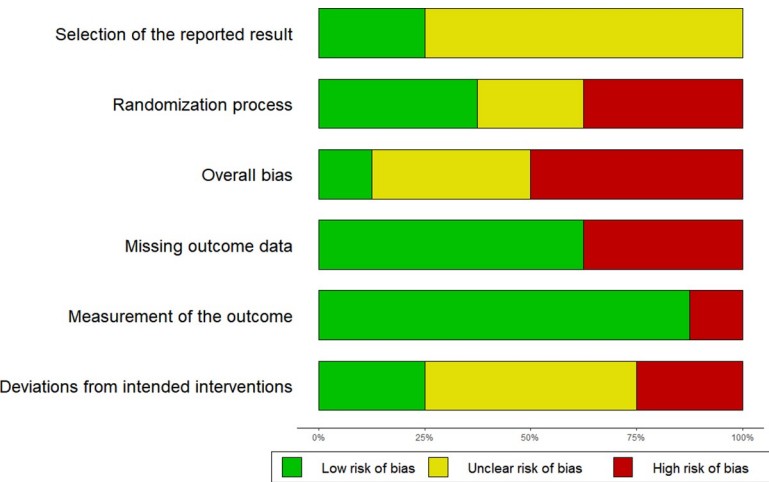

**Fig 3. Risk of bias assessment across the studies included in the meta-analysis by the Cochrane risk of bias tool.**
Represented in the X axis are the percentages of studies for every risk of bias domain.

The outcome domains that did not reach statistical significance were Attention (SMD = 0.282, 95% CI [-0.123, 0.687], p = 0.172), Flexibility (SMD = 0.036, 95% CI [-0.220, 0.294], p = 0.778), General cognition (SMD = 0.130, 95% CI [-0.267, 0.529], p = 0.519), Inhibition (SMD = 0.258, 95% CI [-0.124, 0.641], p = 0.186), Processing speed (SMD = 0.163, 95% CI [-0.091, 0.418], p = 0.207), Reasoning (SMD = 0.161, 95% CI[-0.139, 0.461], p = 0.293), Verbal memory (SMD = 0.305, 95% CI [-0.336, 0.945], p = 0.351) and Visual memory (SMD = 1.370, 95% CI [-0.886, 3.626], p = 0.234).

Due to the low number of studies, $I^2$ was used as the main heterogeneity estimator, since it is not sensitive to the number of trials included. However, the Cochrane's Q statistic yields similar results. High heterogeneity was found in Verbal memory and Visual memory ($I^2 >$ 75%). The cognitive domains that showed a moderate heterogeneity were Attention, Reasoning, Verbal working memory and Visual working memory ($75\% > I^2 > 25\%$). A low heterogeneity was found in Flexibility, General cognition, Inhibition and Processing speed ($I^2 < 25\%$).

Meta-analysis could not be performed on Verbal fluency because it was assessed only by 1 study [29] in which they reported no statistical significance, which is in line with the aggregate in that cognitive function (Hedge's $g$ = 0.04) that yields no effect.

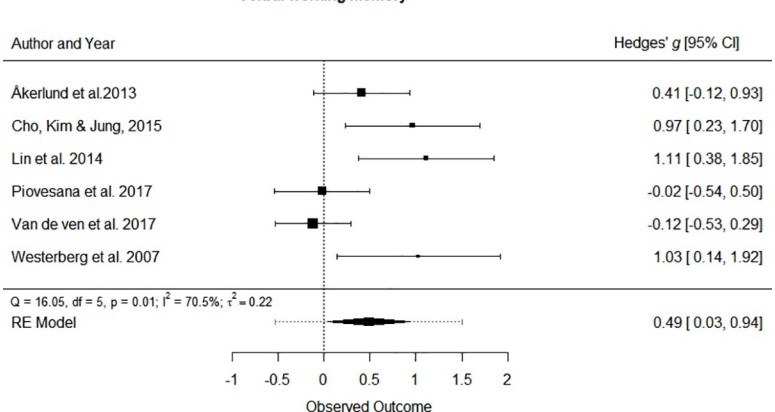

**Fig 4. Forest plot for the Verbal working memory cognitive domain.**

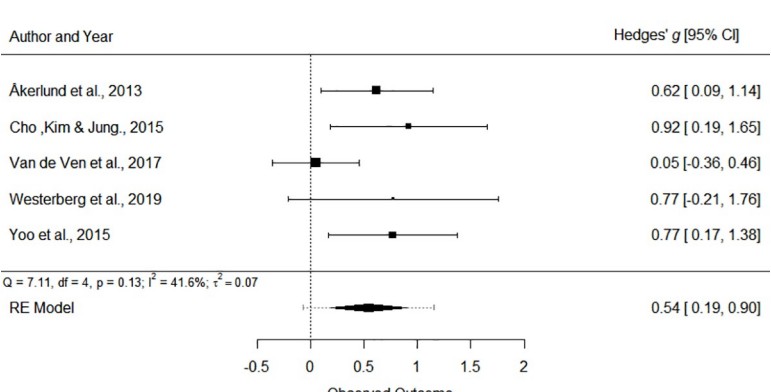

**Fig 5. Forest plot for the Visual working memory cognitive domain.**

## Discussion

The aim of the current study was to assess the effects of computer-based cognitive interventions in the cognitive functioning of ABI patients. To our knowledge, this is the first meta-analysis addressing this topic. The main finding of this study is that Visual and Verbal working memory are improved immediately after computer-based cognitive intervention in ABI population, while other cognitive functions like Attention, Reasoning, Processing speed, Inhibition, Flexibility and Visual and Verbal memory do not seem to improve.

In general, there was a low heterogeneity given the low number of studies included and the difference between the tests used and the population characteristics, which makes the findings more robust.

These findings are in line with previous works in ABI population. Fetta, Starkweather & Jill [13] conducted a systematic review on computer-based cognitive intervention in mild traumatic brain injury population. The authors concluded that there is weak evidence of improvement in working memory after intervention. Working memory is known to be one of the cognitive domains that benefits the more from direct training [35]. However, it has been argued that this improvement lacks generalization to other aspects of cognition and daily living [36, 34], although evidence in computer-based cognitive training from studies that could not be included in the present work due to insufficient data [37, 38], suggest that this training might be beneficial on self-reported occupational performance and dealing with cognitive fatigue.

On the other hand, these results are partially different from the systematic review performed on computer-based cognitive interventions for Attention and Executive functions in ABI [15] population. The authors found that most trials showed a positive effect, which was not replicated in the present work. The main reason might be that when only RCTs are included, the effect on those cognitive domains is non-significant. Virk et al. [39] performed a meta-analysis that found no effect of cognitive remediation in most of attentional deficits after ABI. The main reason of this lack of effect is probably because attention improves better when cognitive remediation is provided together with metacognitive and compensatory strategies [8], while most of the included studies in this work provide cognitive remediation alone.

These results are also in line with a previous systematic review in stroke population [14] that found no evidence of improvement in executive functions after computer-based cognitive intervention. Evidence suggests that metacognitive strategy training and self-regulation is the

best intervention for such deficits, while cognitive training remains a second option [8, 40, 41]. No effect is also found in Verbal and Visual memory, which are better suited for external and internal compensatory strategies and instructional techniques rather than cognitive training [42–44].

Processing speed results are in contrast with the findings of studies done in elder healthy [45] and MCI population [46] that show a significant improvement right after training. This may point that ABI patients are more resistant to processing speed rehabilitation, and the evidence suggests that a compensatory approach such as Time Pressure Management [47] is more suited to this kind of population.

Finally, General cognition might not benefit from computer-based cognitive intervention. However, only 2 effect sizes contributed to that domain, and no standard tests of generalized usage such as Montreal Cognitive Assessment and Mini-Mental State Examination were used in the included studies. Considering that most of the cognitive domains did not benefit from computer-based cognitive intervention, it is to be expected that General cognition does not either, although studies in healthy and MCI show a small improvement on overall cognition [46]. More RCTs using General cognition scales are needed to clarify if General cognition benefits from computer-based cognitive interventions.

## Limitations

The main limitation of the present work is the low number of the included studies. Although the fact that they are RCTs provides robustness to the results, many of the planned analyses in the protocol could not be performed. Meta-regression analyses would have provided valuable information, such as what kind of population benefits more from the intervention (e.g. Stroke vs TBI), what kind of interventions produce better results (e.g. Multi-domain vs Single-Domain), the effects of passive vs active control groups, the effect that study quality has on the final results, the generalization of the cognitive improvements on real-life setting or a transfer effect to other cognitive domains. The intervention characteristics, such as hours of intervention and number of sessions could not be explored as moderators either, which would have provided valuable information since it is has not been addressed in ABI population, with studies suggesting that shorter and less frequent sessions are more effective in healthy people [47] and MCI patients [46]. Funnel plot would have allowed us to examine the publication bias of the studies selected, although given that many of them are have small sample sizes and report non-significant outcomes, it may not be a major issue in this particular topic.

## Conclusions and future directions

Computer-based cognitive interventions that were included in the present work are beneficial for Verbal and Visual working memory immediately after intervention in ABI, although no effect was found in the rest of the cognitive domains addressed. There is a need for more high-quality RCTs investigating possible moderators for a successful rehabilitation. The implementation of long-term and daily living measures is necessary for future trials, since evidence of lasting results and generalization is lacking. Other kinds of cognitive interventions that do not imply direct training were rarely used by the studies included. It would be of great interest to try to combine compensatory strategy training or instructional techniques with computer-based cognitive training, as Man et al. [30] did, to see if there is a bigger effect in the outcomes studied.

Efforts should be made to decrease the risk of bias, since most of the reviewed studies were at risk, especially due to the absence of proper randomization procedures and allocation

concealment. Additionally, pre-registration should be considered to avoid selective reporting of outcomes.

## Supporting information

**S1 Table. PRISMA checklist.**
(DOC)

**S2 Table. Tests used by the included studies to assess the cognitive domains.** When more than one test measured the same function, aggregates were produced for the independent meta-analyses.
(DOCX)

**S1 File. Forest plots and heterogeneity tests of non-significant cognitive domains.**
(DOCX)

## Author Contributions

**Conceptualization:** Rodrigo Fernández López, Adoración Antolí.

**Data curation:** Rodrigo Fernández López.

**Formal analysis:** Rodrigo Fernández López.

**Funding acquisition:** Rodrigo Fernández López, Adoración Antolí.

**Investigation:** Rodrigo Fernández López.

**Methodology:** Rodrigo Fernández López.

**Project administration:** Adoración Antolí.

**Resources:** Adoración Antolí.

**Software:** Rodrigo Fernández López.

**Supervision:** Adoración Antolí.

**Validation:** Adoración Antolí.

**Visualization:** Adoración Antolí.

**Writing – original draft:** Rodrigo Fernández López.

**Writing – review & editing:** Adoración Antolí.

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
