## [Decision Letter · Decision Letter 0]

25 Mar 2020

PONE-D-20-03777

Computer-based cognitive interventions in acquired brain injury: A systematic review and meta-analysis of randomized controlled trials

PLOS ONE

Dear Mr. Fernández López,

Thank you for submitting your manuscript to PLOS ONE. After careful consideration, we feel that it has merit but does not fully meet PLOS ONE’s publication criteria as it currently stands. Therefore, we invite you to submit a revised version of the manuscript that addresses the points raised during the review process.

Reviewers have provided queries on this manuscript with several questions arising. They include a better description of the interventions performed and the main variables considered, and improvement of the introduction and discussion sections. Changes should be applied in text before the acceptation of the work.

We would appreciate receiving your revised manuscript by May 09 2020 11:59PM. To enhance the reproducibility of your results, we recommend that if applicable you deposit your laboratory protocols in protocols.io, where a protocol can be assigned its own identifier (DOI) such that it can be cited independently in the future. For instructions see: http://journals.plos.org/plosone/s/submission-guidelines#loc-laboratory-protocols

We look forward to receiving your revised manuscript.

Kind regards,

Jose A. Muñoz-Moreno, Ph.D.

Academic Editor

PLOS ONE

Journal Requirements:

2. Please ensure that you have discussed how your submission relates to and advances upon the following publication:

https://pubmed.ncbi.nlm.nih.gov/28661947

Reviewers' comments:

Reviewer's Responses to Questions

**Comments to the Author**

1. Is the manuscript technically sound, and do the data support the conclusions?

Reviewer #1: Partly

Reviewer #2: Yes

2. Has the statistical analysis been performed appropriately and rigorously? 

Reviewer #1: I Don't Know

Reviewer #2: Yes

3. Have the authors made all data underlying the findings in their manuscript fully available?

Reviewer #1: Yes

Reviewer #2: Yes

4. Is the manuscript presented in an intelligible fashion and written in standard English?

Reviewer #1: Yes

Reviewer #2: Yes

5. Review Comments to the Author

Reviewer #1: Manuscript #: PONE-D-20-03777

Title: Computer-based cognitive interventions in acquired brain injury: A systematic review and meta-analysis of randomized controlled trials

Thank you for the opportunity to read this manuscript. The purpose of this paper was to conduct a systematic review and metanalysis of RCTs that used computer-based cognitive interventions for persons with acquired brain injury.

Abstract: The abstract overall is clear and concise.

Introduction: The second paragraph of the introduction requires some expansion for the reader. The sentence, “The impact of cognitive rehabilitation has increased over the years…..” needs some further explanation to what that impact is. Is it better for the patient outcomes? Is it being used more?

A definition of the what types of computer-based interventions would also be helpful. There are interventions on the computer that still require a therapist to be involved in the session and make decisions about advancing (such as the Attention Process Training) and there are others that the computer automatically advances the difficulty based on performance. Are all of these included in the review?

Methods:

Regarding the eligibility criteria. Were studies only included if they studied adults or were pediatric populations considered as well?

The search strategy and statistical analysis plan described is adequate for the purposes of this paper.

Results:

The description of the demographics at the top of page 10 should be reworded. “There was heterogeneity in terms of age…..although there was a general consistency in the sex of the participants”. Although does not seem to be the correct word. The authors should note that the higher percentage of males in a TBI population is consistent with other statistics published.

Later in the first paragraph on page 10, the authors describe the various types of trials as multi-domain, working memory………The control groups were usual care, passive and active.” As a reader, I’d like to know more about what these trials mean. Explain the difference between multi-domain and the single domains. What do those interventions look like? Examples would be helpful. The same with the control groups, description of what usual care entailed and what passive and active controls were used would be helpful.

Additional information that was not provided but is necessary includes the setting the RCTs were conducted in. Were these studies in an outpatient setting, acute inpatient rehab, etc? If the studies reviewed reported time since injury to start of intervention that is also an important variable that should be reported.

In the 3rd paragraph on page 10, the authors describe the variation in duration of the interventions provided. Additional information that would be helpful for the reader is the average length for each session and the frequency the sessions were provided.

Discussion and Conclusions: The authors state on page 14 that general cognition may not benefit from cognitive training. This needs to be clarified, do you mean any cognitive training or specifically the computerized cognitive training addressed in this paper? This clarification needs to be addressed throughout the 2nd paragraph on page 14.

Figures and Tables: In Figure 1, the box describing records excluded. The fourth bullet point indicates no intervention in cognitive functions. Should this be no computerized cognitive intervention?

Table 1. For the reporting of age, either a range of standard deviation from each study would be helpful. Also, more information about the interventions such as, length and frequency of each session, where interventions were delivered and were other interventions with a therapist also delivered with these computerized interventions?

Figure 2 More explanation of the figure and what it means is needed.

In general, as a reader of this paper, I wanted to see more explanation of the computerized interventions and how they were delivered. I also wanted to see more information about the outcome measures that were used.

Reviewer #2: The authors conducted a systematic review and meta-analysis of RCTs that assessed Computer-based cognitive interventions in acquired brain injury – which included patients who have suffered a TBI and/or stroke. This is a well written and easy to read manuscript. The tables and flow chart is well done. I have a few items that I would like the authors to address/expand on.

Introduction

1. The authors write “The impact of cognitive rehabilitation has increased over the years (7) and there are considerable efforts in the search of evidence-based interventions (8) for ABI patients.” The authors should briefly expand further on the phrase “impact” – what exactly do they mean by this?

2. The authors write “In order to provide an optimal intervention for every patient, the use of computer-based interventions has increased.” This is an assumption that unless can be backed up, should be re-writteen

3. The authors write “Computer-based cognitive interventions are most commonly provided with the use of tools such as Rehacom and Cogmed, which are based on cognitive training of the different cognitive domains.” I recommend not highlighting any specific company unless the authors can back up that these two companies are factually “most commonly” used.

Discussion

4. The authors write “However, it has been argued that this improvement lacks generalization to other aspects of cognition and daily living (33), although evidence in computer-based cognitive training from studies that could not be included in the present work (34, 35) suggest that this training might be beneficial on daily activities and dealing with cognitive fatigue.” The authors should expand on this (i) Specifically state that none of the reviewed article assessed daily activities and (ii) state why 34 and 35 could not be included

5. The Limitations flow into the ‘suggestions for future research’. This should be separate.

Conclusions

6. The authors conclude “The Computer-based cognitive training is a beneficial intervention for Verbal and Visual working memory after ABI, although no effect was found in the rest of the cognitive domains addressed.” Please rephase this in the context of the studies examined (i) Computer-based cognitive training that --fall within the design parameters summarized here – is a beneficial intervention (ii) benefits occurs ‘immediately’ after intervention.

General

7. There is a 2016 systematic review “Computerized Cognitive Rehabilitation of Attention and Executive Function in Acquired Brain Injury: A Systematic Review by Bogdanova, Cicerone et al. that was not mentioned by the authors. Please appropriately include this review in both the introduction and discussion.

6. PLOS authors have the option to publish the peer review history of their article (what does this mean?). If published, this will include your full peer review and any attached files.

Reviewer #1: No

Reviewer #2: No

---

## [Author Response · Author response to Decision Letter 0]

30 Apr 2020

Response to reviewers

Dear Dr. Jose A. Muñoz-Moreno,

thank you for allowing us to submit a revised draft of our work “Computer-based cognitive interventions in acquired brain injury: A systematic review and meta-analysis of randomized-controlled trials” and for taking the time to review and comment it for improvements. We tried our best to reflect the changes suggested by editor and reviewers. We are grateful to reviewers and editor for their helpful feedback on the work.

Here we response point by point all the points made by reviewers and editor. Please note that when we point the specific location of the changes, we refer to the manuscript with track changes file.

Comments from editor

Editor: “1. Please ensure that your manuscript meets PLOS ONE's style requirements, including those for file naming.”

Response: We found some mistakes in the text style so we changed them according to the requirements. However, we did not find any mistake in file naming after careful examination of the templates. If editor could point out the specific mistake in file naming, we would correct it as soon as possible.

Editor: “Please ensure that you have discussed how your submission relates to and advances upon the following publication: https://pubmed.ncbi.nlm.nih.gov/28661947”

Response: We did as suggested and included this publication in the introduction and discussion. This change can be seen in page 4, second paragraph, line 80-82 and third paragraph of page 16, line 294-297.

Comments from reviewer 1

Reviewer 1:” The second paragraph of the introduction requires some expansion for the reader. The sentence, “The impact of cognitive rehabilitation has increased over the years…..” needs some further explanation to what that impact is. Is it better for the patient outcomes? Is it being used more?”

Response: We agree with reviewer 1 in relation to the word “impact”. It is not clear. We wanted to point that this kind of intervention has gained attention in the literature, so we rephrased it to make it clear, instead of using “impact”. This change can be seen in page 3, second paragraph, line 56.

Reviewer1:” A definition of the what types of computer-based interventions would also be helpful. There are interventions on the computer that still require a therapist to be involved in the session and make decisions about advancing (such as the Attention Process Training) and there are others that the computer automatically advances the difficulty based on performance. Are all of these included in the review?”

Response: Reviewer 1 asked us to provide a definition for the different kinds of computer-based interventions. We think it is a good idea, so we included a brief definition. We included information about the therapist being involved during intervention. We would also like to confirm that all the interventions that Reviewer 1 mentions are included in the review. Changes can be seen in the end of page 3 and the first paragraph of page 4, line 66-75

Reviewer 1: “Regarding the eligibility criteria. Were studies only included if they studied adults or were pediatric populations considered as well?”

Response: We decided not to include age limitation for the analysis as long as subjects do not present neurodevelopmental disorders. The trial with the youngest population included adolescents and children around 10-12 years, so we decided it could be useful for the inclusion.

Reviewer 1: “The description of the demographics at the top of page 10 should be reworded. “There was heterogeneity in terms of age…..although there was a general consistency in the sex of the participants”. Although does not seem to be the correct word. The authors should note that the higher percentage of males in a TBI population is consistent with other statistics published.”

Response: Reviewer 1 make an excellent point. We changed the phrasing to make it clear that these results are in line with previous evidence. This change can be seen in page 12, first paragraph, line 200.

Reviewer 1:” Later in the first paragraph on page 10, the authors describe the various types of trials as multi-domain, working memory………The control groups were usual care, passive and active.” As a reader, I’d like to know more about what these trials mean. Explain the difference between multi-domain and the single domains. What do those interventions look like? Examples would be helpful. The same with the control groups, description of what usual care entailed and what passive and active controls were used would be helpful.

Additional information that was not provided but is necessary includes the setting the RCTs were conducted in. Were these studies in an outpatient setting, acute inpatient rehab, etc? If the studies reviewed reported time since injury to start of intervention that is also an important variable that should be reported.

In the 3rd paragraph on page 10, the authors describe the variation in duration of the interventions provided. Additional information that would be helpful for the reader is the average length for each session and the frequency the sessions were provided.”

Response: We agree with reviewer 1, so we included a brief definition of what multi-domain and single-domain interventions are (end of page 3 and the first paragraph of page 4, line 66-75) and what the different kind of control groups looks like (page 12, second paragraph, line 208-213). We also added information about length and frequency (first paragraph of page 13). Additionally, we decided to include a second table to summarize more information about the RCTs, such as setting, time since injury, frequency and length of intervention (Page 11, Table 2 and page 13, line 229-237).

Reviewer 1: “Discussion and Conclusions: The authors state on page 14 that general cognition may not benefit from cognitive training. This needs to be clarified, do you mean any cognitive training or specifically the computerized cognitive training addressed in this paper? This clarification needs to be addressed throughout the 2nd paragraph on page 14.”

Response: We intended to mean computerized cognitive training, but we can see it is confusing the way it is redacted. We changed it to reflect that it is specifically computerized cognitive training from our results. This change can be found in last paragraph of page 17.

Reviewer 1:”Figures and Tables: In Figure 1, the box describing records excluded. The fourth bullet point indicates no intervention in cognitive functions. Should this be no computerized cognitive intervention?”

Response: We agree. We intended to mean computer-based cognitive intervention, so we changed it to be more accurate. Change can be found in Fig 1.

Reviewer 1: “Table 1. For the reporting of age, either a range of standard deviation from each study would be helpful. Also, more information about the interventions such as, length and frequency of each session, where interventions were delivered and were other interventions with a therapist also delivered with these computerized interventions?”

Response: We agree. We included the standard deviations in the age columns, and we also included an additional table with the required information. Please note that when taking into account standard deviations, mean age differs slightly from the previous ones. Changes can be found in Table 1, Table 2 and first paragraph of page 13.

Reviewer 1:” Figure 2 More explanation of the figure and what it means is needed.”

Response: We added some information to make it clear. However, due to the low number of studies, maybe this figure can be confusing. If reviewers find it not to be useful, we can consider deleting it. The change can be found in line 250-252 of page 14.

Comments from reviewer 2

Reviewer 2: “The authors write “The impact of cognitive rehabilitation has increased over the years (7) and there are considerable efforts in the search of evidence-based interventions (8) for ABI patients.” The authors should briefly expand further on the phrase “impact” – what exactly do they mean by this?”

We agree with Reviewer 2. As we responded to Reviewer 1, the phrasing is misleading and we changed it to reflect that it has gained attention in the literature. This change can be seen in page 3, second paragraph, line 56.

Reviewer 2:”The authors write “In order to provide an optimal intervention for every patient, the use of computer-based interventions has increased.” This is an assumption that unless can be backed up, should be re-written”

Response: We also agree with Reviewer 2 that the phrasing in “In order to provide an optimal intervention for every patient, the use of computer-based interventions has increased” is not optimal, and we changed it to reflect that it has potential but has not proven to be better yet. Change can be found in page 3, second paragraph, line 59-60

Reviewer 2:”The authors write “Computer-based cognitive interventions are most commonly provided with the use of tools such as Rehacom and Cogmed, which are based on cognitive training of the different cognitive domains.” I recommend not highlighting any specific company unless the authors can back up that these two companies are factually “most commonly” used.” 

Response: We agree with Reviewer 2, using the brand names of the programs might not be the best idea, so we changed it to a simple description with no names in it. Change can be found in page 3, second paragraph, line 65-66.

Reviewer 2:” The authors write “However, it has been argued that this improvement lacks generalization to other aspects of cognition and daily living (33), although evidence in computer-based cognitive training from studies that could not be included in the present work (34, 35) suggest that this training might be beneficial on daily activities and dealing with cognitive fatigue.” The authors should expand on this (i) Specifically state that none of the reviewed article assessed daily activities and (ii) state why 34 and 35 could not be included”

Response: We agree that the writing is confusing and can lead to misunderstanding. We re-phrased it to make it clearer and give the information asked and added new information from a trial that was included in the analysis. Change can be found in page 16, second paragraph, line 300-303.

Reviewer 2: “The Limitations flow into the ‘suggestions for future research’. This should be separate.”

Response: We agree, so we separated it as suggested by reviewer 2. This change can be found in the last paragraph of page 18.

Reviewer 2: “The authors conclude “The Computer-based cognitive training is a beneficial intervention for Verbal and Visual working memory after ABI, although no effect was found in the rest of the cognitive domains addressed.” Please rephase this in the context of the studies examined (i) Computer-based cognitive training that --fall within the design parameters summarized here – is a beneficial intervention (ii) benefits occurs ‘immediately’ after intervention.”

Response: We agree that the original phrasing can be misleading. We changed it to reflect what reviewer 2 suggested, being the wording now more accurate and clearer. Changes can be found in last paragraph of page 15, line 288 and first 2 lines of last paragraph of page 18. 

Reviewer 2: “There is a 2016 systematic review “Computerized Cognitive Rehabilitation of Attention and Executive Function in Acquired Brain Injury: A Systematic Review by Bogdanova, Cicerone et al. that was not mentioned by the authors. Please appropriately include this review in both the introduction and discussion.”

Response: Thanks for pointing this out. We modified the introduction and discussion to include the paper suggested and we discussed how it relates to the present work. This change can be found in second paragraph of page 4, line 79-82 and last paragraph of page 16, line 304-308.

Additional clarifications

In the process of introducing the changes suggested by reviewers, we noticed some minor mistakes that we corrected in the manuscript:

• In addition to the studies that we discussed by suggestion of editor and Reviewer 2, we included a systematic review of computer-based cognitive interventions in stroke population by Van de ven et al. (2016) in our work.

• Regarding the hours of intervention and the number of sessions in Piovesana et al (2017) and Westerberg et al (2007), we originally reported a bigger number of hours and sessions than what the actually provided. The reason we made a mistake is because their protocol describes a potential amount of sessions and hours, which is what we originally used for the table. After careful examination, we noticed that they point the actual number of sessions and hours that were provided, which differ from the planned ones. The revised manuscript is already corrected.

• When assessing Risk of bias in Fig 2, we noticed that Van de ven et al. (2017) shows as unclear risk of bias in the randomization process. However, we believe their work has a low risk of bias in that topic. We originally conceived it that way, but we made a mistake when including the information in the excel table. We properly corrected it.

• We want to point out that we updated the funding statement because we got the new included funding recently. We can still assure that the funders had no role in study design, data collection and analysis, decision to publish, or preparation of the manuscript.

We look forward to responding any other question or comments that editor or reviewers may have.

Sincerely,

Rodrigo Fernández López.

---

## [Decision Letter · Decision Letter 1]

17 Jun 2020

Computer-based cognitive interventions in acquired brain injury: A systematic review and meta-analysis of randomized controlled trials

PONE-D-20-03777R1

Dear Dr. Fernández López,

We’re pleased to inform you that your manuscript has been judged scientifically suitable for publication and will be formally accepted for publication once it meets all outstanding technical requirements.

Kind regards,

Jose A. Muñoz-Moreno, Ph.D.

Academic Editor

PLOS ONE

Additional Editor Comments:

We are sorry about the delay in our final decision. COVID-19 crisis is being a hard time for all of us. Reviewers are thankful for the thoughtful responses to their comments and we think the manuscript is convenient now for publication.

---

## [Editor Report · Acceptance letter]

23 Jun 2020

PONE-D-20-03777R1 

Computer-based cognitive interventions in acquired brain injury: A systematic review and meta-analysis of randomized controlled trials 

Dear Dr. Fernández López:

I'm pleased to inform you that your manuscript has been deemed suitable for publication in PLOS ONE. Congratulations! Your manuscript is now with our production department. 

Kind regards, 

on behalf of

Dr. Jose A. Muñoz-Moreno 

Academic Editor

PLOS ONE